# Identification of MicroRNA-Related Tumorigenesis Variants and Genes in the Cancer Genome Atlas (TCGA) Data

**DOI:** 10.3390/genes11090953

**Published:** 2020-08-19

**Authors:** Chang Li, Brian Wu, Han Han, Jeff Zhao, Yongsheng Bai, Xiaoming Liu

**Affiliations:** 1USF Genomics & College of Public Health, University of South Florida, Tampa, FL 33612, USA; lic@usf.edu; 2Huron High School, Ann Arbor, MI 48105, USA; bwus127@gmail.com (B.W.); johnhanhan298@gmail.com (H.H.); 3Marine Military Academy, Harlingen, TX 78550, USA; jeffzhao100@gmail.com; 4Next-Gen Intelligent Science Training, Ann Arbor, MI 48105, USA; 5Department of Biology, Eastern Michigan University, Ypsilanti, MI 48197, USA

**Keywords:** TCGA, microRNA, 3′UTR, cancer, single nucleotide variant, dbNSFP

## Abstract

MicroRNAs (miRNAs) are a class of small non-coding RNA that can down-regulate their targets by selectively binding to the 3′ untranslated region (3′UTR) of most messenger RNAs (mRNAs) in the human genome. Single nucleotide variants (SNVs) located in miRNA target sites (MTS) can disrupt the binding of targeting miRNAs. Anti-correlated miRNA–mRNA pairs between normal and tumor tissues obtained from The Cancer Genome Atlas (TCGA) can reveal important information behind these SNVs on MTS and their associated oncogenesis. In this study, using previously identified anti-correlated miRNA–mRNA pairs in 15 TCGA cancer types and publicly available variant annotation databases, namely dbNSFP (database for nonsynonymous SNPs’ functional predictions) and dbMTS (database of miRNA target site SNVs), we identified multiple functional variants and their gene products that could be associated with various types of cancers. We found two genes from dbMTS and 33 from dbNSFP that passed our stringent filtering criteria (e.g., pathogenicity). Specifically, from dbMTS, we identified 23 candidate genes, two of which (*BMPR1A* and *XIAP*) were associated with diseases that increased the risk of cancer in patients. From dbNSFP, we identified 65 variants located in 33 genes that were likely pathogenic and had a potential causative relationship with cancer. This study provides a novel way of utilizing TCGA data and integrating multiple publicly available databases to explore cancer genomics.

## 1. Introduction

MicroRNAs (miRNA) are small (21–22bp) noncoding RNAs that play a post-transcriptional regulatory role through targeting the 3′ untranslated regions (UTRs) of mRNA(s). MiRNAs regulate mRNA(s) via base pairing through complementary sequences within mRNA molecules. Genetic mutations, especially Single Nucleotide Variants (SNVs) located on the miRNA target site (MTS), can disrupt the binding of targeting miRNAs. This disruption has been reported to be associated with numerous diseases and cancers [1].

Given their functional importance, our understanding of the role of MTS SNVs towards cancer is very limited, and many such SNVs have not even been identified. The Cancer Genome Atlas (TCGA) project provides us with a great opportunity to tackle this problem. TCGA includes sequencing data for 33 cancer types with over 20,000 primary cancer and normal tissue samples. More importantly, it has expression information for mRNAs and miRNAs in both normal and tumor tissues for various cancer types. The anti-correlated miRNA–mRNA pairs in the same tissue between normal and tumor samples could provide us with unprecedented opportunities to investigate tumorigenesis through miRNA regulation and identify functional MTS SNVs associated with a specific cancer type.

Thus, in this study, we utilized such anti-correlated miRNA–mRNA pairs identified in a recent study, [2] to cross-check with publicly available databases—namely dbMTS (database of miRNA target site SNVs) [3] and dbNSFP (database for nonsynonymous SNPs’ functional predictions) [4]—for functionally annotated SNVs. This process validated existing cancer related SNVs and identified and prioritized potential SNVs for future functional and experimental validation. Specifically, our study has identified 23 genes (24 SNVs) from dbMTS and 33 genes (65 SNVs) from dbNSFP that passed our filtering criteria. Cancer association annotation has been conducted on these genes, and our preliminary results reported that several of the identified genes are associated with tumorigenesis.

## 2. Materials and Methods

Anti-correlated expressions of 15 cancer types’ miRNA–mRNA pairs from a previous study [2] were extracted from TCGA (http://cancergenome.nih.gov), and a total of 65,535 miRNA–mRNA pairs were taken as candidates to be used in subsequent analyses. Because miRNA can repress the expression of its targets, we hypothesize that these anti-correlated miRNA–mRNA pairs are likely to be dysregulated in the observed cancer types, which highlight their potential functional importance.

### 2.1. Retrieval and Screening of miRNA Target Site SNVs Using dbMTS

The dbMTS (database of MTS SNVs) is a database containing only single nucleotide variants (SNVs) that could potentially affect the binding of miRNAs. The dbMTS uses computer simulations with multiple popular miRNA target prediction tools to predict the effect of any potential SNV on miRNA. It includes more than 200 annotations for each of its SNVs, to help users assess their functional importance. One piece of important information in dbMTS that we considered was the name of the miRNA–mRNA pair that was disrupted or created by introducing the SNV.

Using our candidate miRNA–mRNA pairs and information of SNVs from dbMTS, we identified all the candidate SNVs in the 3′UTR that could help us select our candidate miRNA–mRNA pairs. A customized Python script was written to retrieve SNVs that could affect binding between the candidate miRNA–mRNA pairs in dbMTS. A brief summary of our method of identifying 3′ UTR SNVs is shown in Figure 1. Specifically, gene symbols in our candidate list were first standardized and converted to Ensembl transcript IDs using Ensembl Biomart (Ensembl Genes 100), as dbMTS includes transcript level predictions. Second, the suffixes ‘-3p’ and ‘-5p’ in dbMTS were ignored and removed, since the miRNAs in our candidate pairs are precursor miRNAs instead of mature miRNAs. Third, using GWAS Catalog [5], GRASP [6], and ClinVar [7], we annotated known cancer-related SNVs using dbMTS. We found 376 variants having at least one annotated cancer-related phenotype. Because a transcript’s 3′ UTR can overlap with coding regions of other transcripts, we then used VEP/Ensembl [8] annotation to remove those SNVs that could potentially reside in coding regions. The VEP/Ensembl annotations for each SNV were retrieved from dbMTS. This process resulted in 24 variants that were exclusively located in the 3′ UTRs of all possible transcripts. Additionally, to investigate if any SNVs identified from dbMTS (a germline SNV database) have been reported as somatic, we checked our identified SNVs that could potentially affect miRNA binding in SomamiR DB 2 [9]. SomamiR DB is a database exclusively designed to study somatic mutations altering miRNA–mRNA interactions. We retrieved somatic SNVs that were located in experimentally identified MTS using CLASH (Cross-linking, ligation, and sequencing of hybrids) or CLIP-seq (Cross-linking immunoprecipitation sequencing) methods.

### 2.2. Retrieval and Screening of Nonsynonymous SNVs

The dbNSFP database includes only nonsynonymous SNVs (SNVs that change the final protein product) in the human genome. It includes more than 300 unique functional annotations for each SNV.

We retrieved all SNVs in dbNSFP that meet the following criteria, as shown in Figure 2:
(1)SNVs located in genes that are included in our candidate gene list;(2)SNVs that include one of the following strings reported for ClinVar phenotype in dbNSFP to ensure their functional relationship with cancer: (‘carcinoma’, ‘sarcoma’, ‘leukemia’, ‘tumor’, ‘cancer’, ‘lymphoma’, ‘myeloma’);(3)FATHMM was used to filter for damaging variants by removing those SNVs that are predicted as ‘Neutral’;(4)SNV allele frequency should not be high (minor allele frequency <0.05 in both 1000Gp3_AF and UK10K_AF) because pathogenic SNVs are not likely to be prevalent in the general population;(5)SNVs that are pathogenic or likely pathogenic as reported by ClinVar clinical significance. These steps were achieved using a custom Python script.


### 2.3. Functional and Pathway Analyses

The Catalogue Of Somatic Mutations In Cancer (COSMIC) [10] database was used to identify the impact of somatic mutations in cancer. To identify somatic mutations in our findings, we downloaded COSMIC mutation data from its targeted and genome wide screens. Our candidate SNVs were checked for their somatic statuses using the variants’ HGVS-nomenclature as keys, and examined for gene–cancer association using COSMIC. The Kyoto Encyclopedia of Genes and Genomes (KEGG) [11] is a database that links gene catalogs to higher-level systemic functions of the cell, the organism, and the ecosystem. We used KEGG exclusively for gene level analysis. Diseases that were found to be associated with our 23 genes were then further analyzed through public websites to find any possible cancer associations. We also employed COSMIC and KEGG for function and pathway validation and conducted similar searches for cancer association of the 33 annotated genes obtained from dbNSFP. These steps of finding functional interpretations of our candidate SNV/gene lists were mainly achieved through manually searching the above-mentioned publicly available databases.

### 2.4. Survival Analyses

Survival analyses were performed using two web-based tools, GEPIA [12] and OncoLnc [13]. Both tools used multivariate Cox regression to campare survival outcomes between patient groups. Kaplan–Meier survival curves were used to visualize the survival differences between patient groups over time. One difference between these two tools is the way they pre-process the TCGA data. To check if this difference affects the outcome, we adopted both tools in the analysis. Our candidate gene list was submitted to these two web-based tools, and corresponding outputs were downloaded and compared. One thing to note regarding these candidate genes is that they would require further experimental studies to mechanically associate their functions to tumorigensis.

## 3. Results and Discussion

### 3.1. Twenty four Non-Coding SNVs were Identified at Potential miRNA Target Sites

As illustrated in Figure 1, we obtained 870,254 SNVs that could potentially affect our candidate miRNA–mRNA pairs after checking with dbMTS. Next, we identified 376 variants with at least one annotated phenotype (GWAS Catalog, GRASP, and/or ClinVar, Appendix A). We identified 24 SNVs exclusively located in the 3′UTR of all available transcripts. These SNVs were predicted by TargetScan to cause either gain of miRNA regulation or substitution of existing miRNA regulation (Appendix A). These candidate SNVs would be overlooked by most of the traditional in-silico functional prioritization tools, because these MTS SNVs are expected to show only moderate effect size to gene expression. This indicates the uniqueness of our approach in identifying cancer type-specific functional SNVs for further validation. These 24 candidate SNVs were in 23 different genes (Table 1), and all of these genes have 2 or more potential cancer associations according to our observed anti-correlated miRNA-gene pairs. We expect that genes involved in multiple cancer types (e.g., *MAFB* and *SLC30A7*) are especially useful in detecting common tumorigenesis pathways for various types of cancer under the influence of similar set of miRNAs. The candidate list of SVNs and genes could account for the cancer association as a result of the anti-correlation between the miRNA–mRNA pairs in the TCGA project, but further experimental and/or clinical validation is required to elucidate their contributions to tumorigenesis.

### 3.2. Two Hundred and Thirty Three Somatic SNVs were Identified in Experimentally Identified miRNA Target Sites

Among the 870,254 SNVs retrieved from dbMTS, we found 233 somatic SNVs in SomamiR DB. Eleven of these SNVs located in MTS were identified by CLASH based experiments (Table 2). These SNVs may be responsible for our observed anti-correlated miRNA–mRNA pairs. For example, gene FLAD1 has been reported to be overexpressed in breast cancer [14], and let-7b dysregulation has been reported to be associated with breast cancer [15]. While no previous study has linked let-7b and FLAD1 together with breast cancer, our analysis shows the possibility of a novel mechanism of tumorigenesis for breast cancer. Additionally, a recent study has linked CDH1 and the miR-30 family with pancreatic cancer [16], which is observed in our study (Table 2). This evidence highlights the ability of our approach to identify informative tumorigenesis SNVs and genes. Additional SNVs located in MTS identified by CLIP-seq-based experiments are provided in Appendix A.

### 3.3. Sixty Five Nonsynonymous SNVs were Identified in the Candidate Gene List

We identified 28,764 variants from the dbNSFP database for all genes from the TCGA analysis results to conduct our filtering process. We utilized three filters in order to identify SNVs that could be associated with cancer. The first filter used was “damaging in the fathmm-XF_coding prediction”; this step kept 12,617 variants. Next, we filtered the variants using The 1000 Genomes Project Phase 3 data by requiring SNVs’ minor allele frequency to be less than 0.05, leaving us with 644 variants. Next, we filtered the variants by requiring minor allele frequency to be less than 0.05 in the UK10K data, leaving us with 139 variants. Lastly, we identified pathogenic or likely pathogenic variants using ClinVar significance, leaving us with 65 candidate variants that could be associated with different types of cancer (Appendix A). 

To check for the somatic status of these 65 identified SNVs, we examined the COSMIC database and identified 31 out of 65 SNVs as somatic, which matched 383 COSMIC records (some SNVs were observed in multiple samples). Interestingly, these identified somatic SNVs were predicted by FATHMM as potentially pathogenic, which is a promising starting point for future functional and clinical studies interested in cancer driver mutations (Appendix A). 

### 3.4. Biological Pathway Analysis

At variant level, among the 24 SNVs identified from dbMTS, we found out that one of the SNVs, rs3044, had been reported to be associated with colorectal cancer [17]. Another SNV, ENST00000467482.5:c.*134C>T, was observed in three patients with lung cancer in COSMIC. The functions of the other SNVs need further biological validation to determine if they can contribute to cancer susceptibility.

At gene level, we obtained 23 different genes that contain the 24 identified variants from dbMTS (gene *XIAP* has 2 variants). We used these 23 genes for cancer and disease association analysis using KEGG pathway annotations. We observed that two diseases are associated with gene *BMPR1A* (Juvenile polyposis syndrome [18], Hereditary mixed polyposis syndrome [19]), which causes an increased risk for colorectal carcinoma. In addition, X-linked lymphoproliferative syndrome is associated with gene *XIAP* [20], which causes lymphomas in approximately one-third of all patients.

Additionally, using KEGG Pathways, we checked 33 different genes containing 65 variants from dbNSFP for cancer and disease association. KEGG results indicated that several of these 33 genes had cancer-related disease associations. Diseases associated with genes *SDHB* and *SDHD* (Cowden Syndrome [21], Malignant paraganglioma [22]), *TSC1* and *TSC2* (Tuberous sclerosis complex [23]), *ERCC6* (Disorders of nucleotide excision repair [24]), *FLCN* (Birt–Hogg–Dube Syndrome [25]), and *CHEK2* (Li-Fraumeni Syndrome [26]) increase risk for various cancer types. Mismatch repair deficiency, associated with gene *EPCAM* [27], and familial adenomatous polyposis, associated with gene *APC* [28], can potentially increase risk for colorectal cancer. Nijmegen syndrome, associated with gene *NBN* [29], increases risk for lymphomas, and basal cell nevus syndrome, associated with gene *PTCH1* [30], increases risk for basal cell carcinoma.

Among the 33 candidate genes identified from dbNSFP analysis, we found that there are nine genes that contain at least one cancer associated variant reported in COSMIC (Table 3). This result indicates that genes harboring multiple variants are likely to be associated with more cancer types, suggesting that some variants are cancer type specific.

### 3.5. Survival Analysis

We conducted a survival analysis of the two disease-associated genes from the pathway analysis, *BMPR1A* and *XIAP*. We first used an online tool, Gepia. For *BMPR1A*, among the three associated cancer types (Table 1), its high expression has significant survival benefits in KIRC (Kidney Renal Clear Cell Carcinoma) (*p* value: 0.00013, Figure 3). For *XIAP*, among the three associated cancer types (Table 1), its low expression has significant survival benefits in BRCA (Breast Invasive Carcinoma) (*p* value: 0.02, Figure 4). We also conducted survival analysis using another online tool, OncoLnc. The results were similar between these two different approaches, which indicate that the candidate genes we identified are potentially functional. 

## 4. Conclusions

Given that two categories of variants (3′UTR and nonsynonymous) are important in changing gene expression, we analyzed the anti-correlated miRNA–mRNA target pairs and target genes obtained from TCGA dataset. We prioritized many candidate SNVs and genes harboring them using dbMTS, dbNSFP, and other functional annotation tools. We believe the disruption resulting from SNV in the miRNA target site is accountable for the observed anti-correlated miRNA-gene pairs, which contributes to tumorigenesis. Additionally, non-synonymous SNVs identified in dbNSFP can effectively change the amino acid sequence, which is also a possible cause for tumorigenesis. The candidate genes and candidate somatic and germline variants identified in this study could provide guidance for wet-lab scientists or clinical researchers in biomarker discovery. However, this will require extensive experimental work or validation from clinical data.

## Figures and Tables

**Figure 1 genes-11-00953-f001:**
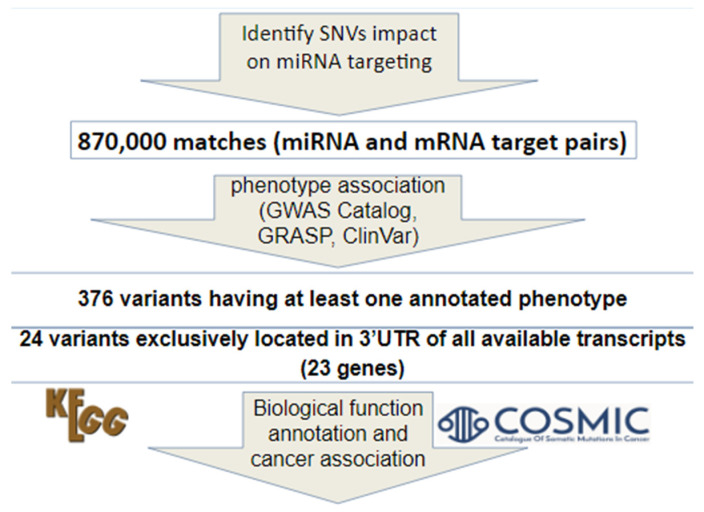
Pipeline for identifying single nucleotide variants (SNVs) using dbMTS and cancer pathway databases.

**Figure 2 genes-11-00953-f002:**
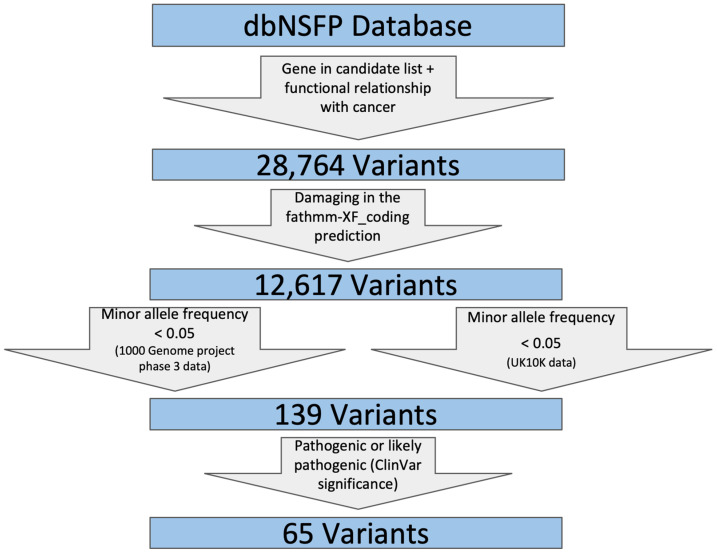
The workflow for filtering nonsynonymous variants using dbNSFP (database for nonsynonymous SNPs’ functional predictions).

**Figure 3 genes-11-00953-f003:**
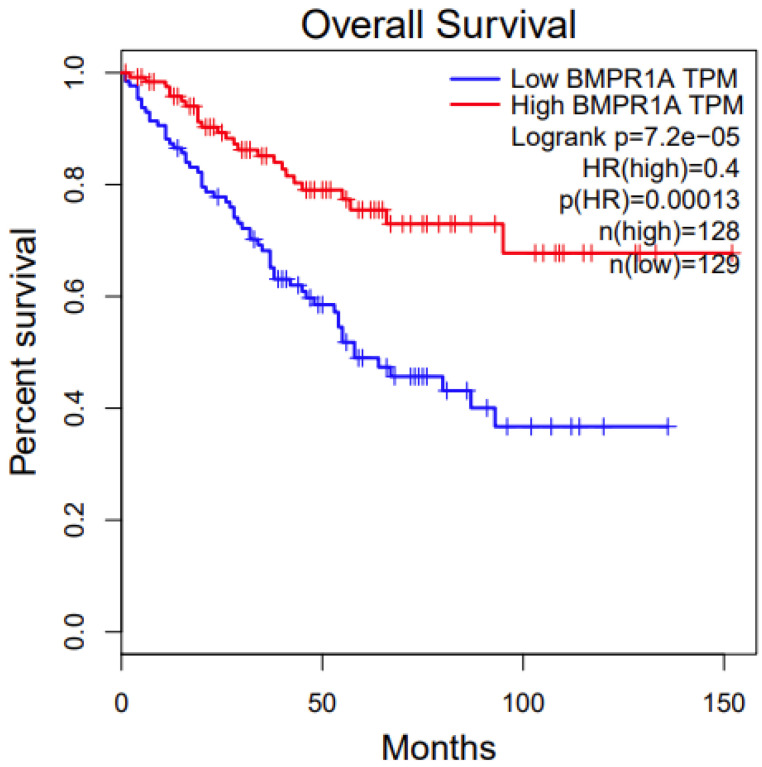
Survival analysis of *BMPR1A* in TCGA (The Cancer Genome Atlas) KIRC (kidney renal clear cell carcinoma).

**Figure 4 genes-11-00953-f004:**
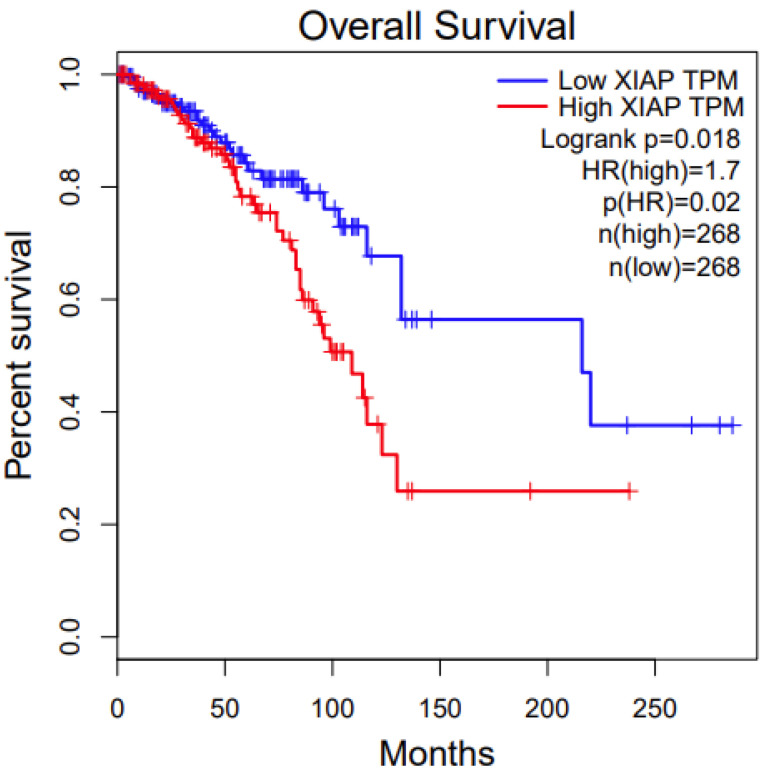
Survival analysis of *XIAP* in TCGA BRCA.

**Table 1 genes-11-00953-t001:** Genes harboring 24 variants and their associated TCGA(The Cancer Genome Atlas) cancer types with disrupted miRNAs.

Gene Symbol	Cancer Type *	Disrupted miRNAs
*ACVR2B*	BRCA, HNSC, STAD	has-miR-4670
*ADAM12*	THCA, BRCA, HNSC, LUAD	has-miR-4726
*BMPR1A*	LUAD, KICH, KIRC	has-miR-520f
*CD93*	BRCA, LIHC, LUAD, LUSC, STAD	has-miR-1179
*CDS1*	BRCA, STAD, UCEC, KIRC, KIRP, LUAD	has-miR-411
*CHST3*	KIRP, LUAD, STAD	has-miR-652
*ELOVL4*	BRCA, UCEC, KIRC	has-miR-185
*FAM20B*	KIRP, LUAD, UCEC, BRCA, STAD	has-miR-649
*FLRT2*	STAD, BRCA, KIRC, LUAD	has-miR-5688
*GIPC2*	LUAD, STAD	has-miR-4765
*GPR143*	KIRC, KIRP, LIHC	has-miR-520f
*HEMK1*	BRCA, STAD, KIRC, KIRP, LUAD	has-miR-3911
*MAFB*	LUAD, PRAD, STAD, BRCA, HNSC, KICH, KIRC	has-miR-1266
*PGR*	LUAD, PRAD, BRCA, KIRC	has-miR-182
*PRKCA*	BRCA, KIRC, KIRP, LUAD, PRAD	has-miR-296
*PTCHD1*	STAD, THCA, BRCA, HNSC	has-miR-185
*SCN2B*	LUAD, PRAD, STAD, BRCA	has-miR-141
*SLC30A7*	LUAD, PRAD, KICH, KIRC, KIRP, STAD, BRCA	has-miR-152
*SLC39A8*	LUAD, LUSC, PRAD	has-miR-141
*TLL2*	LUAD, STAD, BRCA, COAD, KIRC, KIRP	has-miR-29a
*TRAM2*	LUAD, PRAD, STAD, BRCA	has-miR-625
*XIAP*	BRCA, KIRP, LUAD, PRAD	has-miR-580
*ZDHHC15*	STAD, BRCA, KIRC, LUAD, PRAD	has-miR-367

* BRCA: breast invasive carcinoma, COAD: colon adenocarcinoma, HNSC: head and neck squamous cell carcinoma, KICH: kidney chromophobe, KIRC: kidney renal clear cell carcinoma, KIRP: kidney renal papillary cell carcinoma, LIHC: liver hepatocellular carcinoma, LUAD: lung adenocarcinoma, LUSC: lung squamous cell carcinoma, PRAD: prostate adenocarcinoma, STAD: stomach adenocarcinoma, THCA: thyroid carcinoma, UCEC: uterine corpus endometrial carcinoma.

**Table 2 genes-11-00953-t002:** Somatic SNVs located in miRNA target sites (MTS) identified by CLASH (cross-linking, ligation, and sequencing of hybrids) experiments.

Chromosome	Position	Ref. Allele	Alt. Allele	Gene Symbol	MiRNA	Cancer Classification ((Tissue)(Histology))
1	154992969	G	C	*FLAD1*	let-7b	(breast)(carcinoma)
3	49015647	G	A	*WDR6*	miR-222	(large_intestine)(carcinoma)
3	57559757	A	G	*PDE12*	miR-484	(skin)(malignant_melanoma)
5	93593940	G	A	*NR2F1*	miR-149	(oesophagus)(carcinoma)
5	131204977	A	T	*LYRM7*	miR-30e	(liver)(carcinoma)
6	30724942	T	C	*TUBB*	miR-708	(kidney)(neoplasm)
6	43504866	A	T	*TJAP1*	miR-744	(ovary)(neoplasm)
7	56063723	T	C	*CCT6A*	miR-3663-3p	(ovary)(neoplasm)
9	128695535	G	T	*SET*	miR-185	(ovary)(neoplasm)
16	68835430	C	T	*CDH1*	miR-30c	(pancreas)(carcinoma)
19	29675270	C	A	*PLEKHF1*	miR-331-3p	(liver)(carcinoma)

**Table 3 genes-11-00953-t003:** COSMIC results for 65 nonsynonymous variants identified from dbNSFP.

Gene Symbol	Tissue Related to Cancer
*APC*	large intestine, stomach, lung, thyroid, urinary tract
*ATM*	lymphoid, lung, thyroid, large intestine
*CHEK2*	large intestine, urinary tract
*MET*	endometrium, autonomic ganglia, pancreas, lung, large intestine vulva, cervix, kidney, thyroid, ovary, adrenal gland
*NF1*	small intestine
*PTCH1*	ovary, skin
*RET*	thyroid, lung, lymphoid, adrenal gland
*TSC2*	central nervous system, stomach, cervix
*WT1*	skin, kidney, lymphoid

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
