# Peer review of "Identification of MicroRNA-Related Tumorigenesis Variants and Genes in the Cancer Genome Atlas (TCGA) Data"

_genes, 2020, doi:10.3390/genes11090953_

Round 1

Reviewer 1 Report

This paper tries to identify specific variants on MicroRNA that may be related to cancers by cross matching several public databases. The paper is quite concise and in general well-written. It is easy to follow. I found out the research presented in the manuscript is interesting, which tries to study the regulatory effects of microRNA and mRNA target pairs and how variants on microRNA link to cancers. Not many literature can be found extensively studying this problem. Authors presented a data-driven approach to visit this research by exploring available data accumulated at several public databases, and this approach is solid and may provide inspiration for other genomics studies given the large volume of available data. 

It would be better to provide more detailed explanation on the method itself, including how each analytical component was performed, parameters of the software used, scripts, and codes. Currently, it is still very difficult to reproduce the analysis. And the most important contribution is the method this paper presents which may inspire the development of other data-drive analysis methods. 

Reviewer 2 Report

In their manuscript, Liu et al present a computational synthesis of cancer tissue sequencing data and microRNA binding site variant data in order to characterize how functional variants may be correlated to various types of cancer. The role of miRNAs on carcinogenesis on a pan-cancer level has yet to be fully characterized. The analysis presented in this study may be important for the development of new methods to study miRNAs on a large scale. However, there are major revisions and additions that the authors must make in order to increase the validity of their findings.

Major Comments

  1. A primary approach to studying miRNA and gene variants used in this study was identifying cancer-related genes based on data from previous studies. Although referring to previous studies is important, it may be more provocative to correlate specific variants to cancer stage or prognosis in order to make direct correlations between miRNA target site variants and cancer progression. In the conclusion, the authors hypothesize that disruption of the miRNA target sites contributes to tumorigenesis. But there seems to be little primary evidence for this. TCGA houses a wide range of clinical annotations for each patient whose sequencing data has been submitted. Is it feasible for Liu et al to correlate cancer-associated variants of genes and miRNAs to cancer prognosis using these clinical annotations?
  2. In Table 1, Liu et al list genes with 24 SNVs at miRNA target sites and with associations to different cancers. However, the activity of these variants is never explored further, a step that seems important if the reader is to believe in the validity of the miRNA-mRNA pairs being correlated to cancer. For example, ADAM12 has multiple variants across 4 different cancers. However, THCA (depending on the subtype) is much less carcinogenic than BRCA, HNSC, and LUAD. Does the variant of the miRNA binding site affect carcinogenesis? There should be further exploration into the specific variants of such genes and their presence in cancers with differing characteristics. Additionally, the roles of the genes listed in Table 1 should be detailed, in order to clarify to the reader the effects of miRNA-mediated silencing of these genes.
  3. Lines 189-197: What is the significance of using both Gepia and OncoLnc? Do they perform significantly different statistical analyses of survival? What method of survival correlation is used in each? I believe the plots in Figures 3 and 4 use the Kaplan-Meier correlation, but I cannot be sure. The specific correlation types should be identified for both tools used. This information could be detailed in the methods section.
  4. Throughout the manuscript, the authors claim that specific genes are correlated to certain diseases (for example, lines 175-181). I understand that these correlations were made using the KEEG database. But, not citing the studies that contributed to the KEGG database may misinform the reader. It may be more informative to cite the primary literature that has identified the correlations specified.

Minor Comments

  1. There are numerous grammatical and typographical errors throughout the manuscript that affect comprehension at key points. For example, at line 56, the authors state

“Because the repression effect of miRNA to its targeting mRNA, we hypothesize that…”

The authors should carefully check the manuscript for such errors. Additionally, the authors must minimize use of colloquial language.

  1. According to the Genes manuscript template, author contributions must be in the CRediT format (https://img.mdpi.org/data/contributor-role-instruction.pdf).

Round 2

Reviewer 2 Report

I believe that the authors, after specifying a more limited scope of their study, have addressed my concerns adequately. However, it would be beneficial to the reader to more explicitly emphasize that the candidates studied here are not necessarily correlated to tumorigenesis. I suggest that the authors specify in Line 251 that the further research would involve mechanistically associating the candidates found in this study to tumorigenesis. Aside from this small change, I have no further comments.

Author Response

I believe that the authors, after specifying a more limited scope of their study, have addressed my concerns adequately. However, it would be beneficial to the reader to more explicitly emphasize that the candidates studied here are not necessarily correlated to tumorigenesis. I suggest that the authors specify in Line 251 that the further research would involve mechanistically associating the candidates found in this study to tumorigenesis. Aside from this small change, I have no further comments.

Response:

We thank the reviewer for the suggestion. We have clarified this point in Line 251-252 in this revision as suggested.